# Magnitude and modifiers of the weekend effect in hospital admissions: a systematic review and meta-analysis

Yen-Fu Chen,[1] Xavier Armoiry,[1] Caroline Higenbottam,[2] Nicholas Cowley,[3] Ranjna Basra,[4] Samuel Ian Watson,[1] Carolyn Tarrant,[5] Amunpreet Boyal,[6] Elizabeth Sutton,[5] Chia-Wei Wu,[7] Cassie P Aldridge,[6] Amy Gosling,[2] Richard Lilford,[1] Julian Bion[2,6]

For numbered affiliations see end of article.

**Correspondence to**
Dr Yen-Fu Chen;
y-f.chen@warwick.ac.uk

## ABSTRACT

**Objective** To examine the magnitude of the weekend effect, defined as differences in patient outcomes between weekend and weekday hospital admissions, and factors influencing it.

**Design** A systematic review incorporating Bayesian meta-analyses and meta-regression.

**Data sources** We searched seven databases including MEDLINE and EMBASE from January 2000 to April 2015, and updated the MEDLINE search up to November 2017. Eligibility criteria: primary research studies published in peer-reviewed journals of unselected admissions (not focusing on specific conditions) investigating the weekend effect on mortality, adverse events, length of hospital stay (LoS) or patient satisfaction.

**Results** For the systematic review, we included 68 studies (70 articles) covering over 640 million admissions. Of these, two-thirds were conducted in the UK (n=24) or USA (n=22). The pooled odds ratio (OR) for weekend mortality effect across admission types was 1.16 (95% credible interval 1.10 to 1.23). The weekend effect appeared greater for elective (1.70, 1.08 to 2.52) than emergency (1.11, 1.06 to 1.16) or maternity (1.06, 0.89 to 1.29) admissions. Further examination of the literature shows that these estimates are influenced by methodological, clinical and service factors: at weekends, fewer patients are admitted to hospital, those who are admitted are more severely ill and there are differences in care pathways before and after admission. Evidence regarding the weekend effect on adverse events and LoS is weak and inconsistent, and that on patient satisfaction is sparse. The overall quality of evidence for inferring weekend/weekday difference in hospital care quality from the observed weekend effect was rated as 'very low' based on the Grading of Recommendations, Assessment, Development and Evaluations framework.

**Conclusions** The weekend effect is unlikely to have a single cause, or to be a reliable indicator of care quality at weekends. Further work should focus on underlying mechanisms and examine care processes in both hospital and community.

**Prospero registration number** CRD42016036487

## INTRODUCTION

Increased mortality rates among patients admitted to hospital during weekends have received substantial public attention. This

---

### Strengths and limitations of this study

► This systematic review provides a comprehensive summary and appraisal of the international literature published up to November 2017 on the weekend effect associated with mortality, adverse events, hospital length of stay and patient satisfaction.

► The Bayesian meta-analyses take into account variations both within and between studies.

► The review examines different modifiers of the weekend effect using both subgroup analyses of study-level data and subgroup analyses reported within individual studies.

► The review focuses only on hospital-wide sample of admissions and does not include condition-specific admissions.

► Quantitation of the weekend effect does not explain underlying mechanisms.

---

so-called weekend effect has motivated policies to strengthen 7-day services in the UK but has also triggered a heated debate about how to interpret the evidence.[1–4] Hundreds of studies examining the weekend effect in different clinical areas from around the world have now been published, some focusing on unselected emergency admissions, others on elective admissions, and exploring outcomes for specific diagnostic groups.[5–11] More recently, several systematic reviews and meta-analyses have attempted to summarise these studies.[12–14] However, the published reviews have been limited to describing the presence or absence, and estimating the magnitude, of the weekend effect. Few had gone beyond describing the quantitative estimates to explore possible mechanisms behind this apparently ubiquitous phenomenon. In those reviews that attempted to do so, conclusions were drawn from subgroup meta-analyses and meta-regressions of a small number of variables without paying sufficient attention to potential confounding factors

in study-level data and nuanced analyses reported within individual studies.[13] Understanding causation is of crucial importance for healthcare providers, policy makers and patients in order to take actions that are based on an accurate interpretation of the scientific evidence. We have therefore performed a comprehensive mixed methods review of the quantitative and qualitative literature. Here, we report our analysis of the quantitative literature to characterise the magnitude of the weekend effect and explore potential modifiers of the effect.

## METHODS

### Structure of the review

This paper is part of a mixed methods review incorporating a systematic review of the magnitude of the weekend effect and a framework synthesis that examines the underlying mechanisms of the effect. The protocol providing details of the overall study design and methodological approaches has been previously reported.[15] Briefly, the review aims to answer the following overarching question:

What is the magnitude of the weekend effect associated with hospital admission, and what are the likely mechanisms through which differences in structures and processes of care between weekdays and weekends contribute to this effect?

We define the weekend effect as the difference in patient outcomes between weekend and weekday hospital admissions, using the definitions of 'weekend' as those given in the various publications. The research question is addressed through (1) examination of studies providing quantitative estimates of the weekend effect and its possible modifiers and (2) interrogation of diverse (both quantitative and qualitative, primary and secondary) evidence that sheds light on the underlying mechanisms of the weekend effect. The former is reported as a systematic review in this paper, whereas the latter will be described in a companion paper in the form of a framework synthesis. The two components of the mixed methods review shared the same initial comprehensive literature search and study screening process (described below), and were then run in parallel. Review teams of the two component reviews/syntheses shared information with each other on a regular basis, and findings from the two components were used to inform and complement each other.

### Search strategy

Using MEDLINE, CINAHL, HMIC, EMBASE, EThOS, CPCI and the Cochrane Library without language restriction, we limited the search to year 2000 onwards to ensure that evidence reasonably reflected contemporary health organisation and practice. Our iterative search strategy combined terms relating to 'weekend/weekday' or 'out-of-hours' with terms relating to 'hospital admissions'. Terminology used in MEDLINE is shown in online supplementary appendix 1 of the published protocol.[15]

Records were imported into EndNote (Thomson Reuters) and de-duplicated. The initial search in April 2015 was updated with a MEDLINE search in May 2016 and again in November 2017 as our screening of the initial search identified few (1/28) relevant publications uniquely in other databases. We used reference chaining for completeness. Additional searches were undertaken specifically for framework synthesis, described in the companion paper.

### Study selection and eligibility criteria

Records were initially screened by one reviewer. Potentially relevant records were discussed in plenary meetings by both teams to refine study eligibility criteria, and subsequently coded according to the following grouping:

1. Observational studies comparing weekday and weekend admissions with quantitative data on processes and/or outcomes.
2. Studies in which changes in service delivery and organisation at weekends were introduced and the impacts were evaluated quantitatively.
3. Studies providing qualitative evidence that could shed light on the mechanisms of the weekend effect.
4. Studies describing differences in case-mix between weekday and weekend admissions without looking into process of care or patient outcomes.

Studies that fell under (1) above are the focus of this systematic review; studies that were classified into groups (2) to (4) were routed to framework synthesis for further consideration.

A study needed to have met the following criteria to be included in the systematic review:

► Have evaluated undifferentiated admissions to acute hospitals, that is, admissions across different conditions or specialties, rather than being limited solely to those related to specific conditions or specialties. Undifferentiated admissions included emergency and elective adult, paediatric, medical, surgical and obstetric admissions. For studies that reported both aggregated and condition-specific weekend effects, only the aggregated data were used in the quantitative analyses of the systematic review. We chose to focus on unselected, rather than condition-specific admissions to avoid duplicating meta-analyses[8 9 14] focusing on condition-specific admissions.

► Have compared at least one of the following outcomes of interest between weekend admissions and weekday admissions, or between patients having their critical period of care at weekends (eg, receiving a surgical procedure just before weekend; giving birth during weekend) with those having their critical period of care on weekdays: mortality, adverse events (defined as undesirable events caused by medical management rather than the patient's underlying condition), length of hospital stay (LoS) and quantitatively measured patient satisfaction. The definition of 'weekend' and the cut-points for mortality were those given in the various publications.

Studies comparing out-of-hours and regular hours were included if out-of-hours included weekends. We did not study daytime–night-time comparisons alone. We excluded conference abstracts and 'grey literature' because of difficulty assessing risk of bias.

Independent duplicate coding of potentially relevant studies was performed for the first 450 (40%) of potentially relevant records to maximise consistency of approach; the remaining studies were then assessed by single reviewers. Final study selection was determined by two reviewers. Any discrepancies in study coding and selection were resolved by discussions between reviewers or by seeking further opinion from other review team members.

### Data extraction and risk of bias assessment

Data extraction was carried out by one reviewer and checked by another; risk of bias was performed independently by two reviewers. Discrepancies were resolved through discussions. Data from included studies were extracted into a predefined and piloted spreadsheet using a detailed data extraction and coding manual (see online supplementary appendix 1). Information collected included study characteristics, methodological features and quantitative outcomes for weekend and weekday admissions including estimates of the weekend effect and results of sensitivity analyses.

Risk of bias assessment focused on level of statistical adjustment (online supplementary appendix 2). We assigned each study to one of the following four categories, which we developed ad hoc based on emerging evidence on key confounding factors related to hospital mortality[16–20]: 1—comprehensive adjustment; 2—adequate adjustment: 2a—adjusted for measures of acute physiology and 2b—adjusted for contextual factors reflecting the severity or urgency of the patient's condition including route of admission; 3—partial adjustment and 4—inadequate adjustment (see online supplementary appendix 1, p.11).

### Data synthesis

Our prespecified primary outcome is mortality. The quantitative synthesis methods described below were used to analyse mortality data, which form the main part of this article. Data related to adverse events, LoS and patient satisfaction were tabulated and presented in the supplementary file, with a brief narrative summary provided in this article.

### Bayesian meta-analysis

The primary prespecified outcome for the meta-analysis was mortality using the endpoints described in the papers; where multiple mortality endpoints were given, we used mortality at hospital discharge for the main analyses. The data were meta-analysed using a Bayesian random effects model that allowed for within-study variation and between-study heterogeneity (online supplementary appendix 3). Analyses were undertaken using (log) adjusted odds ratios (or hazard ratios or rate ratios if odds ratios were not reported) and the reported standard errors or equivalence. Studies were therefore implicitly weighted by the estimated variance of individual effect estimates. Where multiple estimates based on different reference day(s) were reported, we used the estimate based on or including Wednesday as the reference group. Where the weekend effect was reported separately for Saturday and Sunday, we used the estimate for Sunday in the primary analysis and included both estimates in subgroup and sensitivity analyses (described below). Where different studies appeared to have used data from the same source and period/location (see online supplementary appendix 4), our selection criteria were based on quality of adjustment for potential confounding factors, largest sample size and most up to date.

The primary meta-analysis included all types of admissions. Exploratory subgroup analyses were performed for mixed, emergency, elective and maternity admissions. We calculated the $I^2$ statistic to quantify statistical heterogeneity between studies ($I^2 > 50\%$ indicating a substantial degree of heterogeneity).[21] All statistical models were estimated by Hamiltonian Monte Carlo (HMC) using Stan V.2.16.[22] Four HMC chains were run for 10 000 iterations including 2000 warm-up/burn-in iterations or more iterations in the same proportion if convergence was judged not to have been achieved. Convergence was assessed using visual inspection of trace-plots and the Rhat statistic.

### Exploring potential sources of heterogeneity

We investigated whether the estimated weekend effect is influenced by various factors through a meta-regression, subgroup analyses and sensitivity analyses. Meta-regression allows simultaneous exploration of multiple factors that could influence the magnitude of estimated weekend effects but it is susceptible to confounding. We examined the following variables: study containing emergency admissions (yes/no), containing surgical patients (yes/no), year of data collection (mid-point where multiple years were included), adequacy of case-mix adjustment (as described earlier; reference category was combined 1 and 2a, ie, adjusted for acute physiology). The country effect is specified as a hierarchical random effect.

Subgroup meta-analyses were performed by types of admissions as described above, and we summarised additional subgroup analyses within individual studies. Sensitivity analyses that we were able to perform were limited because of insufficient data and heterogeneity between studies, increasing the risk of confounding. We focused on including or excluding studies with partially overlapping data, and examining evidence within individual studies (eg, where a study reported both in-hospital and 30-day mortality) to determine the potential impact of methodological differences on the estimated weekend effect.

## Assessment of publication bias

We constructed funnel plots to assess 'small study effects' (studies of smaller sample sizes tend to report larger estimated effects), for which publication bias and outcome reporting bias are among the possible causes.[23] Where funnel plot asymmetry was observed, we used a data augmentation approach to derive a pooled estimator assuming the asymmetry was caused by publication bias.[24]

## Assessment of overall quality of evidence

We followed the Grading of Recommendations, Assessment, Development and Evaluations (GRADE) framework to rate the overall quality of evidence for each of the four outcomes examined in this review. Based on this framework, evidence from observational studies starts with a baseline quality rating of 'low'. The rating for each outcome is then downgraded or upgraded according to our assessment against each of the eight criteria (risk of bias, inconsistency, indirectness, imprecision and publication for potential downgrading[25]; large magnitude of effect, dose response and direction of effect of plausible confounding factors for potential upgrading).[25 26]

## Patient and public involvement

Patients and the public were not involved in the design and conduct of this systematic review, which focuses on published literature. The HiSLAC project, which funded this review, received advice from patient and public representatives through their memberships in the Project Management Committee.

## RESULTS

### Literature search and study selection

After removing duplicates, 6441 records were retrieved and screened, 613 of which passed through first-stage screening. Of these, 224 were routed to framework synthesis and 319 were excluded (see flow diagram in online supplementary appendix 5). Sixty-eight studies (reported in 70 articles) met our inclusion criteria. Altogether, these studies included over 640 million admissions (with some overlap between studies).

### Characteristics of included studies

Key characteristics of the selected 68 studies are shown in online supplementary appendix 6. Studies were predominantly from North America (USA n=22, Canada n=4) and Europe (UK n=24, Ireland n=3, Denmark n=2, Netherlands n=2, Italy n=1, Spain n=1). One study included data from four countries (Australia, Netherlands, UK and USA).[27] Hospital admissions occurred between 1985 and 2016. Sample sizes of individual studies ranged from 824 admissions from a single hospital[28] to 3 51 170 803 admissions from a nationwide database.[29] Patient populations included all types of admissions (11 studies[20 27 29–37]), all medical admissions (1 study[38]), all surgical admissions (3 studies[39–41]), emergency admissions (22 mixed,[16–18 42–60] 6 medical,[19 28 61–66] 6 surgical[67–72]), elective surgery (5[73–77])

and maternity admissions (13[78–90]). Two studies focused on paediatric patients.[31 69] All studies were retrospective cohort studies except one,[37] which was based on comparison of responses to cross-sectional surveys between patients admitted at weekends and those admitted during weekdays. The majority of studies (62/68) used data obtained from administrative databases mostly maintained by national health services (eg, Hospital Episode Statistics from the UK NHS), government affiliated institutions (eg, databases under the Healthcare Cost and Utilization Project run by the Agency for Healthcare Research and Quality) or individual hospitals. Only one study included a dataset from a private insurer.[32] Fifty-six studies evaluated mortality outcomes of various definitions, 19 examined adverse events and 15 assessed LoS.

### Risk of bias

Only one study[16] was considered to have adjusted comprehensively for potential confounding factors including measures both of acute physiology (haematology and biochemistry test results) and admission source (referral by general practitioners or through the emergency department (ED)). Adjustment was considered adequate in three small studies (two of which came from the same hospital[61 65] and the other included four hospitals[19]) through inclusion of measures of acute physiology in regression models, and in 17 studies through adjustment of contextual factors reflecting the acuity/urgency of the patient's conditions in addition to other major confounders. Twenty studies were rated as achieving partial adjustment and 27 studies inadequate adjustment. The rating of statistical adjustment for individual studies can be found in the table for characteristics of included studies in online supplementary appendix 6.

### Mortality

Fifty-six of the included studies examined various mortality outcomes (eight of which focused on neonatal mortality).

#### Bayesian meta-analysis

Results of planned analyses are presented below. All estimated models show good convergence of the chain. HMC trace-plots for the primary analysis, and Rhat statistic and effective sample sizes for all meta-analyses can be found in online supplementary appendix 7.

#### Overall summary estimate

Bayesian meta-analysis including all types of admissions (with minimal overlapping data) is shown in figure 1. The pooled estimate suggested that weekend admissions are associated with a 16% (95% credible interval (CrI) 10% to 23%) increase in the odds of death compared with weekday admissions.

The estimated weekend effect varies widely between individual studies and between sub-populations within studies. The $I^2$ for heterogeneity (which measures between-study variance relative to total variance) appears low but this was estimated with substantial uncertainty

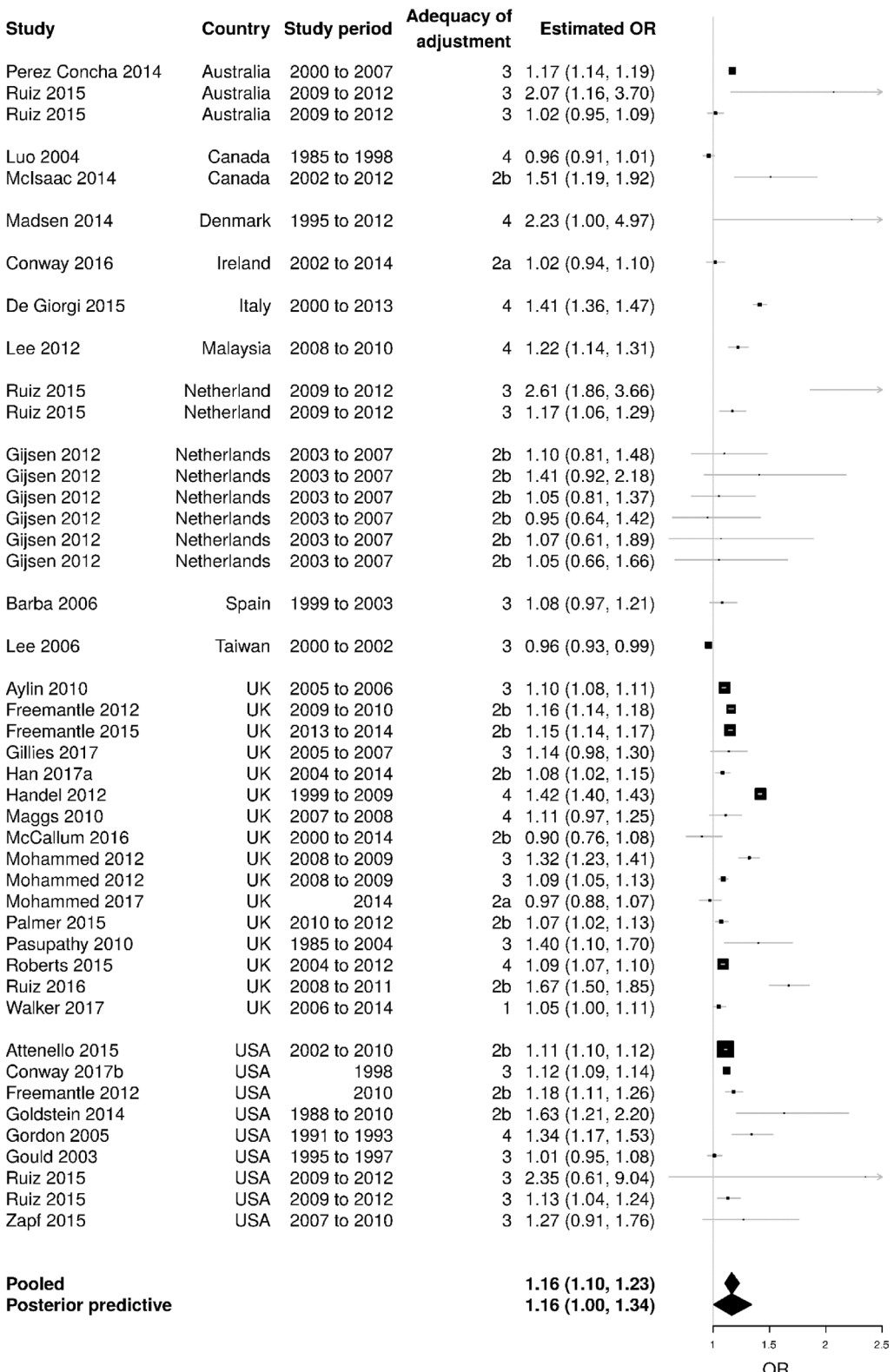

**Figure 1** Bayesian meta-analysis covering all types of admissions for the weekend effect on mortality (sorted by country). Note: Mohammed 2012[35] and Ruiz 2015[27] contributed to two estimates for each country as the weekend effect was estimated separately for different sub-populations (eg, emergency and elective admissions). 'Posterior predictive' indicates the predictive interval (see main text) obtained from the Bayesian meta-analysis. $I^2$=16% (95% CrI for $I^2$ 0% to 62%). The $I^2$ represents the ratio of between-study variance to total variance in this three-level model. The apparently low $I^2$ could be attributed to the between-study variance being relatively small compared with the between-estimate variance within individual studies. As the wide CrI indicates, the $I^2$ was estimated with substantial uncertainty. Several studies included in the review were not included in this meta-analysis due to substantial overlap of data between studies; in this case, studies that were judged to have adopted the most comprehensive statistical adjustment were selected. CrI, credible interval.

**Table 1** Results of meta-regression models of the weekend effect on mortality

| Parameter | Number of estimates in category | Estimate (95% CrI) | % difference in odds ratio (compared with baseline/ reference category) (95% CrI) |
|---|---|---|---|
| Intercept | – | 0.05 (–0.10, 0.20) | (Baseline/reference category OR) 1.05 (0.90, 1.22) |
| Adequacy of statistical adjustment | | | |
| 1 or 2a: Adjustment including measures of acute physiology | 5 | Reference | Reference |
| 2b: Adequate adjustment of main and contextual factors | 40 | 0.13 (–0.03, 0.30) | 14% (–3% , 35%) |
| 3: Partial adjustment | 40 | 0.13 (–0.03, 0.29) | 14% (–3%, 34%) |
| 4: Inadequate adjustment | 34 | 0.15 (–0.01, 0.31) | 16% (–1%, 37%) |
| Surgical admissions yes | 81 | –0.04 (–0.14, 0.06) | –4% (–13%, 6%) |
| Elective admissions yes | 27 | 0.27 (0.21, 0.32) | 31% (24%, 38%) |
| Maternity admissions yes | 23 | –0.18 (–0.26,–0.10) | –17% (–23%, –10%) |
| Time (linear trend) | 119 | 0.00 (0.00, 0.00) | 0% (0%, 0%) |
| Total number of observations/ estimates | 119 | | |

Time (year) was selected as mid-point of the data collection period. Categories 1 (comprehensive adjustment) and 2a (adequate adjustment including measures of acute physiology) were combined due to the low number of studies in these categories. Estimates can be interpreted as approximate percentage increase in the estimate of weekend effect OR. Meta-regressions also have country random effect (varying intercept for countries). Individual studies can contribute to multiple estimates of the weekend effect, for example, by individual years, different patient subgroups and individual weekdays/weekend days (eg, Saturday vs Wednesday and Sunday vs Wednesday). CrI, credible interval.

(16%, 95% CrI 0% to 62%). Posterior predictive interval suggests that if a new study were to be undertaken, the estimated OR is likely to lie somewhere between 1 (no weekend effect) and 1.34 (the odds of death being 34% higher for weekend admissions compared with weekday admissions).

*Sensitivity analysis*
Sensitivity analysis allowing for some overlapping of data between studies produced a result (OR 1.15, 95% CrI 1.10 to 1.22) that is very similar to the main analysis (OR 1.16, 95% CrI 1.10 to 1.23). Funnel plot for the main meta-analysis showed some level of asymmetry and notable statistical heterogeneity between studies of large sizes (online supplementary appendix 8). Use of data augmentation methods (that assume funnel plot asymmetry was caused by publication bias and 'adjusting' for its effect) reduces the estimated weekend effect (OR 1.11, 95% CrI 1.08 to 1.13, online supplementary appendix 8).

Meta-regression
Results from multivariate meta-regression are shown in table 1. The main findings are as follows:

1. Studies that included measures of acute physiology in their statistical adjustment (adequacy of statistical adjustment group 1 or 2a) tended to produce an estimate of the weekend effect that is closer to null and on average reported estimates that are approximately 15% lower in terms of increased odds of mortality compared with studies without adjusting for these measures (groups 2b, 3 and 4).

2. The weekend effect is significantly larger for elective admissions compared with emergency admissions, and significantly smaller (or does not exist) for maternity admissions.

3. There is no apparent time trend in the weekend effect. However, this does not necessarily agree with assessment of time trend within individual studies (see the next section).

4. The above findings need to be interpreted with caution. For example, the finding regarding statistical adjustment relies on data from five estimates reported in four relatively small studies[16 19 61 65] that adjusted for measures of acute physiology, and the 95% CrIs include zero (table 1). Therefore, there is still a substantial level of uncertainty, and the apparent effect of adjustment of acute physiology could have been confounded by other patient or service features associated with the availability of these measures.

Exploring the sources of heterogeneity
Meta-regression allows simultaneous exploration of multiple factors that could influence the magnitude of estimated weekend effects using study-level variables, but its statistical power is limited and is susceptible to confounding by study level variables. This subsection presents findings from additional subgroup and

**Table 2** Subgroup analyses of the weekend effect on mortality by types of admissions

| Analysis | N | Pooled mean (95% CrI) | Posterior predictive mean (95% CrI) | I² (95% CrI) |
|---|---|---|---|---|
| All admissions* | 18 | 1.13 (1.09, 1.18) | 1.13 (1.04, 1.22) | 0.19 (0.00, 0.74) |
| Emergency admissions | 32 | 1.11 (1.06, 1.16) | 1.11 (0.94, 1.31) | 0.44 (0.00, 0.90) |
| Elective admissions | 12 | 1.70 (1.08, 2.52) | 1.70 (0.64, 4.11) | 0.44 (0.00, 0.93) |
| Maternity admissions | 6 | 1.06 (0.89, 1.29) | 1.06 (0.75, 1.53) | 0.44 (0.00, 0.96) |

*This analysis focuses on best adjusted studies that include mixed (both emergency and elective admissions within the same study, with or without including maternity admissions); it thus differs from the main Bayesian meta-analysis (pooled mean 1.16, 1.04 to 1.23) which, in addition to studies included in this meta-analysis, also includes individual types or sub-types of admissions provided that they do not overlap with studies that cover mixed types of admissions.
CrI, credible interval; N, number of observations (estimates of the weekend effect from individual studies).

sensitivity analyses, paying particular attention to within-study comparisons to explore in more detail potential modifiers of the weekend effect.

### Weekend effects by types of admission

Subgroup meta-analyses by types of admissions are summarised in table 2, and individual forest plots are presented in online supplementary appendix 9.1. The weekend effect was observed across different types of admissions, with a potential exception of maternity admissions. Heterogeneity is high within individual types of admissions, indicating the involvement of other factors. Within-study comparisons show that the weekend effect is greater for elective than for emergency admissions (online supplementary appendix 9.1, p.47), confirming the finding from meta-regression.

Among emergency admissions, one study from England[17] and another from the USA[20] demonstrated that the observed weekend effect was largely attributable to 'direct' admissions from the community (eg, general practitioner or walk-in clinic referrals) rather than those through the ED. Another US study restricted to admissions through the ED[57] also showed a substantially smaller weekend effect compared with other studies including all emergency admissions (online supplementary appendix 9, p.48).

### Weekend effect by time period and country

Although meta-regression showed no indication that the weekend effect has changed over time, analyses within individual studies showed a more varied picture (online supplementary appendix 9.2). No time period effects were observed in studies using various databases in the UK, but a significant reduction in the weekend effect over time was reported in a large US study of emergency admissions based on the National Inpatient Sample,[43] and a small study of emergency medical admissions in a single Irish hospital.[63] Within each admission type, variation in the reported weekend effect is apparent among studies from different countries (online supplementary appendix 9.1 and 9.3); however, standardised data allowing cross-country comparisons are very limited.[27]

### Weekend effects by disease condition

Several studies provided subgroup analyses of the weekend effect based on the main diagnostic category related to the admission. The weekend effect was consistently found in admissions associated with conditions such as aortic aneurysm, pulmonary embolism and cancer, and was absent for admissions associated with conditions such as chronic airway obstruction; evidence on the presence of the weekend effect was less consistent for conditions such as myocardial infarction and intracerebral haemorrhage (online supplementary appendix 9.4). In the only study that was judged to have achieved comprehensive statistical adjustment,[16] the test for interaction showed no significant difference (p=0.86) in the estimated weekend effects between admissions associated with different conditions based on the Clinical Classification Software groups.

### Correlation of hospital weekend effect with staffing level

Two studies have attempted to correlate measures of weekend staffing (for consultants)[42] and/or weekend services[52] with observed weekend effect and/or changes in the weekend effect over time for individual hospitals in England. Neither showed an appreciable correlation (online supplementary appendix 9.5).

### Influence of statistical adjustment

Statistical adjustment was carried out in most studies in an attempt to account for different characteristics between weekday and weekend admissions. The number and nature of variables included in statistical adjustment varied widely between studies.

Only six publications reporting studies from a small number of individual hospitals or hospital groups have included measures of acute physiology in the statistical adjustment.[16 19 61–63 65] One of the studies[16] included all emergency admissions while the remaining focused on emergency medical admissions. The weekend effect was substantially diminished by adjustment for severity. Adjustment using measures of acute physiology appears to be sensitive to completeness of data and other factors (online supplementary appendix 10, p.56).

### Influence of other methodological features

Included studies used different definitions of the weekend; most defined the weekend as Saturday and Sunday (n=28) or referred to 'weekend' without defining the term (n=14). Others used various cut-off times in Friday evening or Saturday morning as the starting time and in Sunday evening or Monday morning as the end time of the weekend (n=19). Seven studies included Friday daytime admissions in the weekend group.[33 40 61 68 74 77 79] Different studies also used different measures of mortality in terms of timing (eg, 7 day, 30 day) and place (in-hospital or any location) of death, and different effect measures (eg, odds ratios and hazard ratios). These methodological variations do not usually result in dramatic changes in findings within individual studies, but are likely to have contributed to the statistical heterogeneity between different studies (online supplementary appendix 10, p.56–59).

### Adverse events

Nineteen studies compared the risk of adverse events between weekend and weekday admissions.[28 29 31 39 41 56 69 72 74 77–80 84 85 88–91] While some reported an increased risk for weekend admissions, overall, the findings were heterogeneous across different adverse events within individual types of admissions, and the existence and magnitude of a weekend effect linked to a given adverse event were often inconsistent (online supplementary appendix 11). None of the studies adjusted for physiological severity of illness: sicker patients (and particularly non-survivors) are more susceptible to adverse events.[92]

### Length of stay

Fifteen studies compared hospital LoS between weekend and weekday admissions (online supplementary appendix 12).[19 28 33–35 43 62 63 65–67 69 70 72 74 89] The majority of studies show that the (unadjusted) mean or median hospital LoS was shorter (by 1 day or less in most cases) for admissions during weekends compared with admissions during weekdays, with a few exceptions among studies including elective and maternity admissions.[33–35 74 89] The shorter LoS associated with weekend admissions appears to be partly attributable to the higher proportion of patients who died in the hospital among weekend admissions.

### Patient satisfaction

One study based on data from the 2014 English NHS adult inpatient survey reported a significantly higher level of satisfaction in the information given to them in the ED for patients admitted through ED at weekends compared with those admitted through this route on weekdays.[37] After adjustment for potential confounders, no significant differences between weekend and weekday admissions were found in other domains covered by the inpatient survey (online supplementary appendix 13).

### GRADE assessment of overall quality of evidence

The overall quality of evidence was rated as 'very low' for each of the outcomes (mortality, adverse events, LoS and patient satisfaction) examined in this review primarily due to the observational nature of evidence and inadequate or complete lack of adjustment for potential confounding factors in the majority of included studies. Further details on the GRADE assessment are presented in online supplementary appendix 14.

## DISCUSSION

This systematic review of studies reporting the weekend effect in broad ranges of admissions to hospital has found that weekend admission is associated with a 16% increase in the risk of death, but the magnitude of the effect varies by different types of admissions, case-mix and illness severity, geographic location and contextual and methodological factors.

The overall estimate of the weekend effect varies in meta-analyses published to date, for example, a pooled adjusted odds ratio of 1.12 (95% CI 1.07 to 1.18) by Hoshijima et al,[12] 1.11 (95% CI 1.10 to 1.13) by Zhou et al[14] and a pooled relative risk of 1.19 (95% CI 1.14 to 1.23) by Pauls et al.[13] Our meta-analysis covers by far the largest number of admissions; our pooled adjusted OR of 1.16 (95% CrI 1.10 to 1.23) is broadly in line with other studies, whereas the wider CrI may, in part, reflect the use of Bayesian methods which appropriately account for both within-study and between-study variations. Each of the above meta-analyses covers at least tens of millions of admissions, and yet the estimated weekend effects could differ by nearly twofold. A clear message is that such an estimate is subject to a large amount of noise due to the myriad of contextual factors and different underlying mechanisms associated with different studies and admissions, which need to be examined more closely—and this is the key contribution of our review.

Weekend admissions differ from weekdays: fewer patients are admitted at weekends despite similar weekend–weekday ED attendance rates (thus creating a reduction in the denominator of the weekend mortality ratio),[17] and those who are admitted are sicker (case-mix).[16 19 65] There is scant evidence to support the contention that hospital care is of inferior quality at weekends: adverse events may be more common but confounding by illness severity has not been excluded. In stroke care, different patterns of variation in timeliness and adherence to best practice standards have been reported across the week, with no difference in weekend and weekday admission mortality rates.[93] In one study, vital signs were recorded more reliably at weekends than on weekdays.[19] The finding that weekend mortality effect is larger among elective than emergency admissions might be explained by a greater case-mix difference between weekends and weekdays that is unaccounted for by statistical adjustment among elective admissions compared with emergency admissions. For example, for procedures that are often carried out during elective admissions, such as hip and knee replacement and surgery for large bowel,[75] switching the timing of admission from weekdays to weekends due to change

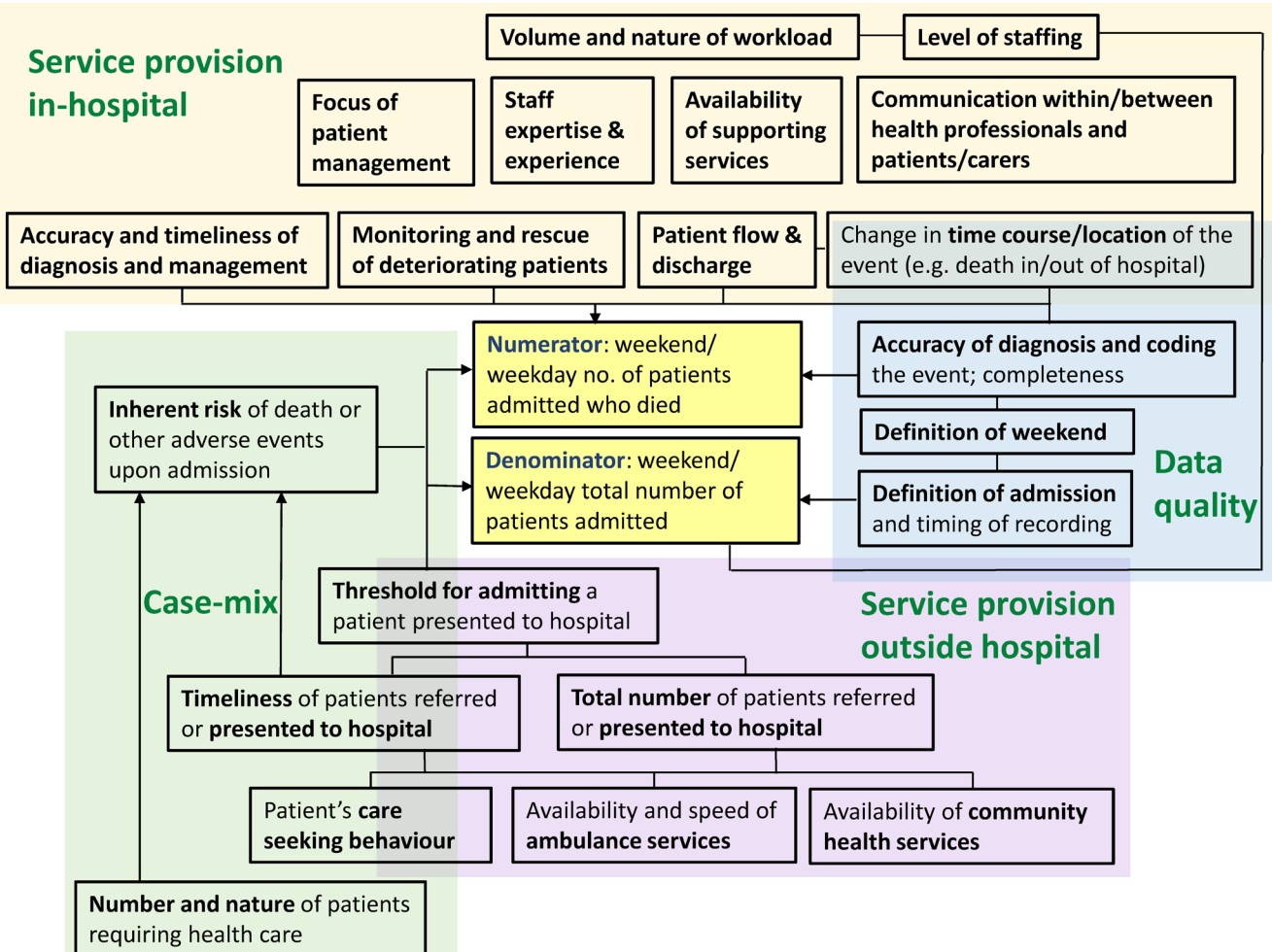

**Figure 2** Factors that may contribute to or modify the weekend effect.

in urgency (which is unlikely to be captured by administrative database) or delay in admission during weekday due to capacity issues ('overflow') is fairly plausible. On the other hand, a greater weekend effect associated with elective admissions is also consistent with the hypothesis that hospitals are configured to care for emergencies at weekends, while elective admissions might be overlooked.

Our review clearly illustrates the old wisdom that large volumes and advanced statistical techniques cannot make up for the inherent limitation within the data. Our assessment of overall quality of evidence using the GRADE framework reinforces the need to appreciate the weakness in available evidence when using observed weekend effect to make an inference on quality of hospital care at weekends. Nonetheless, careful examination of the data may help pin point areas for further investigation. For example, our findings show that the observed weekend effect is substantially larger among elective admissions compared with emergency admissions. Identifying specific types of elective admissions associated with the most profound weekend effect could point to patient pathways and clinical processes that warrant close examination or intervention.

Determining the proximate causes for the weekend effect requires a detailed study of the whole care pathways including health service provision, care processes and patient experience in the community, at the interface between community and hospital, and in hospital following admission on weekdays and at weekends. The paucity of published literature on quantitatively measured patient satisfaction is surprising,[37] as patient's, carer's, and service provider's experience must be at the centre of the design and delivery of health services. We will fill in these important evidence gaps through our companion framework synthesis, and other components of the HiSLAC project.[94–97]

While our estimation of the overall association between weekend hospital admission and mortality is broadly in line with those reported previously,[12–14] our review has several unique strengths. First, previous reviews have either examined only mortality,[12–14] or mortality with a small number of care process or outcome measures for specific disease conditions.[6 9] Our review covers institution-wide and/or nationwide samples of hospital admissions and examined adverse events, LoS and patient satisfaction, in addition to death. Second, previous reviews

have focused on using study-level data to generate pooled estimates of the weekend effect. We have extended this by examining the more nuanced analyses available within individual studies.

This systematic review was limited by the exclusion of condition-specific admissions, although others have extensively reviewed these separately. Nevertheless, we believe the limitation is not a major threat for the validity of our conclusions as we have carefully triangulated the findings by examining subgroups both across and within studies and by carrying out sensitivity analyses. We have attempted to focus on more recent evidence by restricting our inclusion to studies published from year 2000 onwards. However, some of the included studies (14/68) covered admissions prior to 2000. This is unlikely to have substantial impacts on our findings as our meta-regression did not identify a significant time trend. Due to resource constraint (and paucity of data in the case of patient satisfaction), we were unable to carry out more sophisticated analyses for non-mortality outcomes.

Most studies included in this review use routinely collected administrative data. Our review suggests the need for caution in the analysis and interpretation of these information sources. For example, data on important confounders such as severity of illness are often unavailable, and undiscriminating adjustment of other variables such as hospital teaching status and bed size could risk 'adjusting away' some of the weekend effect attributable to care quality. Differential data quality between weekend and weekday admissions is another potential contributor to the weekend effect.[19 29] We recommend a shift of focus from final adjusted mortality rates to considering how different pathway factors influence these estimates (figure 2), using configurative analyses (pattern identification) to supplement aggregative (pooled) approaches.[98]

## CONCLUSION

Weekend admissions are associated with a 16% increase in the risk of mortality. However, the overall quality of evidence is very low. Increasing evidence suggests that the weekend effect on mortality may be largely attributable to case-mix and contextual factors surrounding admissions, and therefore the cause may lie upstream of the care pathway, in the community. In addition, the magnitude of estimated weekend effect can be influenced by methodological approaches and data quality. These suggest that the weekend effect is not a good measure of care quality in hospitals at weekends. Future research and interpretation of research findings on the weekend effect must go beyond the narrow focus of case-mix adjustment of routine hospital data and attempt to examine the broader issues related to the whole care pathway both within and outside the hospital; the quality and availability of data that can allow measurement of care quality with minimal bias; and importantly, take into account the experience of patients, carers and care providers.

**Author affiliations**

$^1$Division of Health Sciences, Warwick Medical School, University of Warwick, Coventry, UK

$^2$University Hospitals Birmingham NHS Foundation Trust, Birmingham, UK

$^3$Worcestershire Acute Hospitals NHS Trust, Worcester, Worcestershire, UK

$^4$University Hospitals of North Midlands NHS Trust, Stoke-on-Trent, Staffordshire, UK

$^5$Department of Health Sciences, University of Leicester, Leicester, UK

$^6$University Department of Anaesthesia & Critical Care, Institute of Clinical Sciences, University of Birmingham, Birmingham, UK

$^7$National Taiwan University Hospital, Taipei, Taiwan

**Acknowledgements** We thank study authors who kindly responded to our queries and peer reviewers for their comments which helped to improve our manuscript.

**Contributors** Y-FC led the preparation of the review, contributed to all stages from conceptualisation to drafting the manuscript of the review and is the guarantor; XA contributed to all stages (except data analysis) of the review; CH, NC and RB contributed to data extraction and quality assessment; SIW planned and carried out Bayesian analyses; AB contributed to literature search, study screening and coordination; CT and ES contributed to development of review methods and study screening; C-WW contributed to data checking; CPA contributed to development of review methods and managerial support; AG contributed to study coding; RL contributed to development of review methods and provided senior advice; JB contributed to development of review methods, provided senior advice and is the principal investigator for the HiSLAC project. All authors commented on draft manuscripts and approved the final version.

**Funding** The High-intensity Specialist Led Acute Care (HiSLAC) project is funded by the NIHR Health Services and Delivery Research (HS&DR) Programme (Project No. 12/128/17). Y-FC, SIW and RL are supported by the NIHR Collaboration for Leadership in Applied Health Research and Care (CLAHRC) West Midlands.

**Competing interests** None declared. The views and opinions expressed herein are those of the authors and do not necessarily reflect those of the HS&DR Programme, NIHR, National Health Services or the Department of Health.

**Patient consent for publication** Not required.

**Provenance and peer review** Not commissioned; externally peer reviewed.

**Data sharing statement** All data relevant to the study are included in the article or uploaded as supplementary information.

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
