## [Reviewer comments · BMJ Open]

ARTICLE DETAILS

TITLE (PROVISIONAL)	The magnitude and modifiers of the weekend effect in hospital admissions: a systematic review and meta-analysis
AUTHORS	Chen, Yen-Fu; Armoiry, Xavier; Higenbottam, Caroline; Cowley, Nicholas; Basra, Rajna; Watson, Samuel; Tarrant, Carolyn; Boyal, Amunpreet; Sutton, Elizabeth; Wu, Chia-Wei; Aldridge, Cassie; Gosling, Amy; Lilford, Richard; Bion, Julian

VERSION 1 - REVIEW

REVIEWER	Elizabeth N Kuhn University of Alabama at Birmingham
REVIEW RETURNED	21-Aug-2018

GENERAL COMMENTS	I must strongly commend the authors for tackling such a nuanced, yet critical, topic. The sheer volume of work required to produce this manuscript is laudable. It adds important information to the literature, though some revisions will strengthen its contribution. The main limitation of the manuscript in its current form, is inadequate adjustment based on 1) sample size and 2) quality of evidence. This manuscript should be reviewed by a statistician qualified in Bayesian methods. The authors stratify studies by the degree to which they adjust for potential confounding factors. How did the authors come up with the categories for comparability of the cohorts on the basis of (design or) analysis? My understanding of the Newcastle Ottawa Scale (discussed in Appendix 2) is that there are only three categories—correction for primary factor, correction for any additional factor, and no correction. The information depicted in Appendix 6 is very important and should be included as a table within the main manuscript (as opposed to within the appendices). Within this table, you should add a column showing the Comparability rating. Were studies weighted based on sample size and/or quality of evidence (in this case Comparability rating)? Please elaborate on this. It is not explicitly stated that all studies were retrospective cohort studies (though I assume they were).
---

	Page 13, line 21 states that hospital admissions occurred between 1985 and 2016. However, in the methods, lit review was restricted to those studies published from 2000 onward in order to reflect modern healthcare practices. Those patients who were admitted from 1985 (>30 years ago) should probably be excluded. Ideally, admission in or after 2000 should be an inclusion requirement. Page 13, Line 32. Administrative databases were used by the majority of studies. Were these databases maintained by the hospital, government, insurance company, independent party? Do we know that these databases included ALL admissions? For example, some administrative outcomes databases include only a random sample of admissions/patients. Page 6, Line 16 – What are the most common definitions of “weekend”? Adverse events are defined as “undesirable events caused by medical management rather than the patient’s underlying condition.” Can those two things really be differentiated? Page 14, Lines 41-44. How do you reconcile that “overall the level of heterogeneity is low” while the estimated weekend effect varies significantly between individual studies. Page 17, line 51. Capitalize “In” Page 47, line 14. The middle column states “Select ONE option from a) to d)” but the options are numbered 1-4. Please correct. Additionally, in the right column, in section 2a), the definition refers to “factors listed in b) and c)”. Similarly in section 3), there is reference to “factors listed in a) and b)”, and in section 4), there is reference to “factors listed in c)”. Please correct to ensure consistency. In all tables and figures, please provide a legend for all abbreviations.
--	--

REVIEWER	David Barer Newcastle University, UK
REVIEW RETURNED	24-Aug-2018

GENERAL COMMENTS	In analysing data from 68 studies, including a total of 640M hospital admissions, this must surely be the biggest health care study ever reported (although over half the cases were from one national database study). Though important, the topic is sadly of much greater political than clinical interest, so hopefully this definitive meta-analysis (and its companion study) will be the last word in the current debate about the evidence “for and against” the weekend mortality effect! Naïve clinicians may assume that analysis of such enormous datasets with multiple variables can “adjust away” all sources of confounding, exposing the comparison of interest in its naked form. This false impression can be reinforced by including a highly diverse case-mix (eg from elective minor procedures to life-threatening emergencies) so that a model incorporating simple administrative data can “account for” the lion’s share of the mortality variation. The present authors are very clear about the
---

limitations of adjusting for case-mix using administrative data alone, and the need to include more direct measures of illness severity.

The efficiency of illness severity adjustment is bound to be more limited in studies of unselected admissions, however, and indeed the authors judge “comprehensive statistical adjustment” to have been achieved in only one of the studies reviewed (Walker et al, Lancet 2017). This study used simple blood tests as indicators of physiological stress, and these accounted for 1/3-1/2 of the excess mortality between weekend and midweek days. In studies of specific diseases, severity indicators are potentially more efficient (for instance, differences in initial neurological score between weekend and weekday acute stroke admissions accounted for the whole of the apparent “weekend mortality effect” in one hospital <https://doi.org/10.1093/ageing/afw173>)

Thus, Chen et al have carried out a comprehensive systematic review and sophisticated, computer-intensive statistical analysis of studies with obvious methodological limitations. I am not qualified to judge all the technical details, but the Bayesian meta-analysis approach provides the flexibility to investigate various cofactors, subgroups and secondary outcomes, included in some studies but not others, analysing trends both within and between studies. This results in a convenient format for clinical readers, but the main conclusions read like those of a narrative review and it is sometimes difficult to track back to check the robustness of the evidence on which they are based. The Bayesian approach does highlight the many sources of mortality variation other than the weekend effect, however, and I strongly agree with their conclusion that future research should focus on some of these, especially where they may be modifiable (although I personally do not find complex organisational flow diagrams like Fig 2 particularly helpful).

From the background and technical information given in the extensive appendices, it is clear that great care was taken in the design and implementation of the study. Standard meta-analytic approaches, such as the “risk of bias” assessment for individual studies, had to be modified and methods developed to account for data overlap between studies.

Some of the subgroup comparisons are interesting, though the labelling of Figs 3-5 is not clear (apparent duplicate studies not explained). The inter-study variation in the ratio of numbers of weekend and weekday admissions and in the actual mortality rates shown in Table 7 is interesting

The apparently large weekend effect seen for elective admissions is bizarre and deserves fuller discussion. Among which type of elective admissions did the main mortality occur and what were the actual death rates? If they were admitted for procedures, were these done at weekends and when did the deaths occur? Were planned admissions for palliative care included? Surely this must be a case-mix (denominator) effect rather than reflecting any deficiencies in care – why would any hospital plan to bring patients in on days when it cannot provide adequate cover?

I have no substantial criticisms of this study – its only weaknesses are in the individual studies selected (systematically) for review. Minor changes might be made to the layout to help the reader (eg the order of topics in the Appendix differs from that in the Results section of the paper) and the labelling of some of the figures and tables could be clearer.

Although the authors emphasise that it is dangerous to draw conclusions from this mass of data about quality or optimal

	organisation of care, I feel they miss an opportunity to go further in their recommendations or future research. Instead of struggling to find ever more intricate methods of case-mix adjustment, why not mine the data to find plausible instances of deficiencies in care in specific conditions or situations, and possible ways in which they might be remedied? The more sophisticated case-mix, process and outcome data now becoming available routinely could then be used to mount cluster randomised trials (at minimal cost), which would eliminate much of the confounding that bedevils current inter-hospital and time trend comparisons. This study is a good illustration of the fact that, even by analysing huge numbers of cases, using sophisticated statistical and intensive computational techniques to reduce noise in the data, we will never produce a LIGO detector for health care effects... 1 typo on p21, ln7 –“two-folds” (this sentence should be shortened)
--	---

REVIEWER	Toshiya Shiga International University of Health and Welfare, Japan
REVIEW RETURNED	25-Aug-2018

GENERAL COMMENTS	Very comprehensive review with the elegant statistic methods. I would like to mention the only one thing. The authors need to show the level of evidence with GRADE (Grading of Recommendations, Assessment, Development and Evaluation) system on each outcome (in this case, mortality, length of stay, or stuff like that)
---

REVIEWER	Jacob Simmering University of Iowa, United States
REVIEW RETURNED	10-Oct-2018

GENERAL COMMENTS	Chen and co-authors present a systematic review and meta-analysis of the literature regarding the "weekend effect" among inpatient admissions. They report finding an increased risk of inpatient mortality among those admitted on the weekend relative to other days of the week. They do not find any additional risk for adverse events for patients admitted during the weekend. The average and median length of stay among weekend admissions was somewhat (less than 1 day) shorter than during weekday admissions - an effect attributed to greater risk of inpatient mortality. In general, the review and meta-analysis seem complete and statistically well-grounded. I have some limited concerns about the authors interpretations of the regression model and about the possible sources of the weekend effect. My full comments are below. Statistical Analysis The use of Bayesian modeling is well-suited to this meta-analysis and provides easy to understand results in the form of posterior probabilities and a well-specified model. I have some brief comments here:  1. The convergence was assessed by visual inspection of the MCMC traces and the $R_{\hat{}}$ statistic. I would like to see, at least in an appendix, the MCMC traces and I would like to see the
--

R_hat statistics and effective sample sizes (n_eff) reported in the main paper. These are important to assessing the quality of the convergence and the extent to which the posterior space has been sampled. Without being confident that the chains are well-mixed and converged, it is hard to assess whether the inference based on the sampling is valid.

2. The MCMC was only run for 2,000 iterations. While convergence, especially in a relatively straightforward model can be achieved in such few runs, I would prefer to see a larger number of iterations (perhaps 10,000 or more) to be sure that any low-probability area in the posterior density is explored.

3. What was the number of iterations discarded as burn-in? What is the number of iterations used for inference?

4. The authors select weak priors but opt for a normal distribution (or half-normal distribution). Can you motivate these priors over alternatives (such as $U(0, \infty)$ or $U(-\infty, \infty)$)? I don't think this would change the results - the selected standard deviations are \gg the estimated size of the effect. It would be nice to provide some basis for selecting the standard normal priors based the a priori estimated size of the parameter.

Statistical Interpretation

1. On page 15, the authors report that the estimated weekend effect diminished with increased statistical control. Specifically, they reported that studies with "adequate adjustment of main and contextual factors" had a 14% greater estimated weekend effect than studies with "adjustment including measures of acute physiology." I am concerned by this interpretation are no further dose-response pattern is observed. If we are to understand the difference between studies with control levels of 1/2a and 2b as being caused by insufficient statistical control, we would expect a similar effect moving to partial adjustment (3) and inadequate adjustment (4). However, we do not observe such an effect. I am concerned that the observed difference between control levels 1/2a may have more to do with study design (e.g., access to labs data) than with the adequacy of statistical control.

2. Moreover, the reported effects are not statistically significant and the reference group strata is sparse ($n = 5$). The authors acknowledge this in point 4 on page 15; however, this is not in agreement with the initial claims in point 1. Point 1 seems to substantially overstate the evidence provided by this analysis.

Major Comments

1. Figure 2 seems unsupported by the text. This perhaps belongs in the sister paper. This paper seems simply about the presence and size of the weekend effect and does not seem to explore the possible causes. Likewise, the sentence in the conclusion that the weekend effect "may be largely attributable to case mix and contextual factors surrounding admissions, and therefore the cause may lie upstream of the care pathway, in the community" seems unsupported by the analysis and review presented.

2. While it can be argued that the weekend effect is not a measure of quality of care (while mortality is increased, adverse events are not suggesting the observed increase in mortality may be the result of case-mix changes and not changes in care quality), this

	paper does not seem to strongly argue that position. This section of the results and conclusion seem to overstate the strength of the evidence presented in the paper. 3. The discussion of the companion paper seems out-of-place in this paper. This paper's focus, as I understand it, is a review and meta-analysis of the literature regarding the weekend effect on mortality, adverse events and length-of-stay. The companion paper explores possible causes of this effect. It is unclear why the companion paper is being discussed in this review. Minor Comments 1. The Figure 1 includes the posterior predictive estimate and credible interval; however, this value is never referenced in the text. Please discuss this value in the text. 2. The paper's hierarchy is somewhat hard to follow. The transition from sensitivity analyses to the results on adverse effects/LOS/patient satisfaction was somewhat surprising. The headings do not indicate hierarchy well or there are too many levels of hierarchy.
--	--

VERSION 1 – AUTHOR RESPONSE

Reviewer 1: Elizabeth N Kuhn, University of Alabama at Birmingham		
I must strongly commend the authors for tackling such a nuanced, yet critical, topic. The sheer volume of work required to produce this manuscript is laudable. It adds important information to the literature, though some revisions will strengthen its contribution. The main limitation of the manuscript in its current form, is inadequate adjustment based on 1) sample size and 2) quality of evidence.	We thank the reviewer for the comments. We address the points concerning adjustment based on sample size and quality of evidence in our responses below.	-
This manuscript should be reviewed by a statistician qualified in Bayesian methods	We believe this has been fulfilled by Reviewer 4.	-
The authors stratify studies by the degree to which they adjust for potential confounding factors. How did the authors come up with the categories for comparability of the cohorts on the basis of (design or) analysis? My understanding of the Newcastle Ottawa Scale (discussed in Appendix 2) is that there are only three categories— correction for primary factor, correction for any additional factor, and no correction.	As the reviewer rightly pointed out, the item concerning comparability of cohorts in the Newcastle-Ottawa Scale allows only three categories (correction for the primary factor, correction of any additional factor, and no correction). From our past experience and pilot of the Scale for this review, the categorisation and wording of the item would not allow differentiation of different levels of statistical adjustment given the large	

	number of potential confounders that could influence hospital mortality - it was difficult to pre-specify a single primary factor, and the vast majority of studies would have adjusted for more than two important factors such as age and main diagnosis and thus be given the highest rating of “two stars” for this item in the original scale. We therefore pre-specified a three-category classification with different wording in our review protocol, and further refined the classification into four categories during the review. We have revised the text to make it clear that the categories were developed by us for this review: “We assigned each study into one of the following four categories, which we developed ad hoc based on emerging evidence on key confounding factors related to hospital mortality.”	Manuscript, p.10
The information depicted in Appendix 6 is very important and should be included as a table within the main manuscript (as opposed to within the appendices). Within this table, you should add a column showing the Comparability rating.	Acknowledging its importance, Appendix 6 (Characteristics of included study table) is a fairly large table spanning seven pages. Our past experience suggests that the journal may prefer to keep such a large table in an appendix / online supplementary material. We would therefore seek the editor’s guidance on this before we move the table into the main manuscript. The comparability rating for each study is already shown in the first column of the table.	Supplementary file, p. 29-35
Were studies weighted based on sample size and/or quality of evidence (in this case Comparability rating)? Please elaborate on this. It is not explicitly stated that all studies were retrospective cohort studies (though I assume they were).	The pooled estimate was informed (through the model) by reported variance of individual effect estimates, which in turn was influenced by sample sizes of individual studies. Therefore studies (or more precisely, effect estimates reported in studies) were implicitly weighted by sample sizes. We have revised the texts of Data synthesis section to read: “Analyses were undertaken using (log) adjusted odds ratios (or hazard ratios or rate ratios if odds ratios were not reported) and the reported standard errors or equivalence. Studies were therefore implicitly	Manuscript p.11

	weighted by the estimated variance of individual effect estimates.” We did not directly apply weighting based on comparability rating because we were aware that the rating is likely to be a fairly imprecise measure of evidence quality (see our response to Reviewer 4’s comment on statistical interpretation below), and there is no accepted consensus with respect to what relative weights should be attached to evidence of different quality rating. Nevertheless, evidence quality (based on our comparability rating) was taken into account in our analyses to some degree: in situations where multiple studies have produced estimates using overlapping data/admissions, we chose estimates from studies with the best comparability rating to be included in our analyses. All studies were retrospective cohort studies except one which was based on cross-sectional surveys. We have added this fact to the description of “Characteristics of included studies”.	Manuscript, p.14
Page 13, line 21 states that hospital admissions occurred between 1985 and 2016. However, in the methods, lit review was restricted to those studies published from 2000 onward in order to reflect modern healthcare practices. Those patients who were admitted from 1985 (>30 years ago) should probably be excluded. Ideally, admission in or after 2000 should be an inclusion requirement.	Studies included in the review vary substantially with regard to the periods within which the admissions took place. The study (Pasupathy et al. 2010) which includes admissions dating back to 1985, for example, covers the period between 1985 and 2004 without providing analyses for different years/periods. To avoid throwing out potentially relevant evidence by setting a strict cut-off criterion on year of admission (the cut-off would also have been an arbitrary choice), we have adhered to the pragmatic criterion based on publication year. We examined time period effect both in our meta-regression (manuscript p.15) and in subgroup analyses (supplementary file Appendix 8.2). Overall no significant time trend was found, and therefore the inclusion of studies with	Manuscript, p.25-6

	pre-2000 admissions is unlikely to have major impacts on our findings. We have added the following texts to our Discussions to address this point: “We have attempted to focus on more recent evidence by restricting our inclusion to studies published from year 2000 onwards. However some of the included studies (14/68) covered admissions pre 2000. This is unlikely to have substantial impacts on our findings as our meta-regression did not identify a significant time trend.”	
Page 13, Line 32. Administrative databases were used by the majority of studies. Were these databases maintained by the hospital, government, insurance company, independent party? Do we know that these databases included ALL admissions? For example, some administrative outcomes databases include only a random sample of admissions/patients.	The databases were mostly maintained by national health services such as the UK NHS, government affiliated institutions such as the Agency for Healthcare Research and Quality (AHRQ) and individual hospitals. Only one study included a dataset from a private insurer. We have now included the information in the text under “Characteristics of included studies”. We are aware that some of the databases (such as those falling under the AHRQ’s Healthcare Cost and Utilization Project) only include random samples of admissions, but do not perceive this to be an issue for estimating the weekend effect.	Manuscript p.14
Page 6, Line 16 – What are the most common definitions of “weekend”?	We have now included the following texts in the Results: “Included studies used different definitions of the weekend; most defined the weekend as Saturday and Sunday (n=28) or referred to “weekend” without defining the term (n=14). Others used various cut-offs in Friday evening or Saturday morning as the starting time and in Sunday evening or Monday morning as the end time of the weekend (n=19). Seven studies included Friday daytime admissions in the weekend group.”	Manuscript p.21
Adverse events are defined as “undesirable events caused by medical management rather than the patient’s	We acknowledge that it is sometimes difficult to make a distinction between undesirable events caused by medical management and those caused by the	-

underlying condition.” Can those two things really be differentiated?	patient’s underlying condition. Nevertheless, this definition of adverse events is widely adopted in health services research literature, including studies included in this review. It is unlikely that the definition would have introduced bias for the estimation of weekend effect on adverse events given the retrospective design of the included studies – adverse events were recorded by people who were unaware of the subsequent intention to make comparison between weekday and weekend admissions.	
Page 14, Lines 41-44. How do you reconcile that “overall the level of heterogeneity is low” while the estimated weekend effect varies significantly between individual studies.	Thank you for pointing out this inconsistency in our original statements. We have reflected upon the findings and can offer the following explanation: We observed an I² value of 16% in the primary meta-analysis, which is generally considered low. However two factors need to be taken into account when interpreting this statistic: (1) Given the multilevel nature of our model, we were able to work out variance at three different levels: the variance of individual estimate of the weekend effect, the variance between different estimates within individual studies, and then the variance between individual studies. A low I² in this case is likely to reflect a lower level of between-study variance relative to variance between different estimates within individual studies. (2) The I² itself has a very wide credible interval and thus is unreliable. We have therefore modified our main text, and have also highlighted the two factors that need to be taken into account when interpreting the findings in the footnote of the Figure.	Manuscript p.16 & Figure 1
Page 17, line 51. Capitalize “In”	This has now been corrected.	Manuscript p.20
Page 47, line 14. The middle column states “Select ONE option from a) to d)” but the options are numbered 1-4. Please correct. Additionally, in the right column, in section 2a), the definition refers to “factors listed in b) and c)”.	We thank the referee for spotting the errors. They have now been corrected.	Supplementary file p.11

Similarly in section 3), there is reference to “factors listed in a) and b)”, and in section 4), there is reference to “factors listed in c)”. Please correct to ensure consistency.		
In all tables and figures, please provide a legend for all abbreviations.	We have now provided a legend for all abbreviations in footnotes of all tables and figures.	Tables and figures throughout the manuscript and appendices in the supplementary file.
Reviewer 2: David Barer, Newcastle University, UK		
In analysing data from 68 studies, including a total of 640M hospital admissions, this must surely be the biggest health care study ever reported (although over half the cases were from one national database study). Though important, the topic is sadly of much greater political than clinical interest, so hopefully this definitive meta-analysis (and its companion study) will be the last word in the current debate about the evidence “for and against” the weekend mortality effect!	Thank you.	-
Naïve clinicians may assume that analysis of such enormous datasets with multiple variables can “adjust away” all sources of confounding, exposing the comparison of interest in its naked form. This false impression can be reinforced by including a highly diverse case-mix (eg from elective minor procedures to life-threatening emergencies) so that a model incorporating simple administrative data can “account for” the lion’s share of the mortality variation. The present authors are very clear about the limitations of adjusting for case-mix using administrative data alone, and the need to include more direct measures of illness severity.	We agree entirely that this is an important issue, and it is one of the key messages that we would like to get cross.	-
The efficiency of illness severity adjustment is bound to be more limited in studies of unselected admissions, however, and indeed the authors judge “comprehensive statistical adjustment” to have been achieved in only one of the studies reviewed (Walker et al,	We agree with the reviewer and have added the following texts to emphasise these points: “Our review clearly illustrates the old wisdom that large volumes and advanced statistical techniques	Manuscript p.24

Lancet 2017). This study used simple blood tests as indicators of physiological stress, and these accounted for 1/3-1/2 of the excess mortality between weekend and midweek days. In studies of specific diseases, severity indicators are potentially more efficient (for instance, differences in initial neurological score between weekend and weekday acute stroke admissions accounted for the whole of the apparent “weekend mortality effect” in one hospital https://doi.org/10.1093/ageing/afw173) Thus, Chen et al have carried out a comprehensive systematic review and sophisticated, computer-intensive statistical analysis of studies with obvious methodological limitations. I am not qualified to judge all the technical details, but the Bayesian meta-analysis approach provides the flexibility to investigate various cofactors, subgroups and secondary outcomes, included in some studies but not others, analysing trends both within and between studies. This results in a convenient format for clinical readers, but the main conclusions read like those of a narrative review and it is sometimes difficult to track back to check the robustness of the evidence on which they are based. The Bayesian approach does highlight the many sources of mortality variation other than the weekend effect, however, and I strongly agree with their conclusion that future research should focus on some of these, especially where they may be modifiable (although I personally do not find complex organisational flow diagrams like Fig 2 particularly helpful).	cannot make up for the inherent limitation within the data. Nonetheless, careful examination of the data may help pin point areas for further investigation. For example, our findings show that the observed weekend effect is substantially larger among elective admissions compared with emergency admissions. Identifying specific types of elective admissions associated with the most profound weekend effect could point to patient pathways and clinical processes which warrant close examination or intervention.” Given the size of the review, we have to be very concise when drawing up the main conclusions. We have clearly presented the numerical estimates from our primary meta-analysis and key subgroup analyses on mortality. For other outcomes and additional analyses, we feel narrative summaries may be a more suitable format to convey our message and avoid putting undue emphasis on numerical estimates given the caveats concerning data and other methodological factors that we highlighted. We believe Figure 2 is useful for highlighting the many factors that could influence hospital mortality and confound the weekend effect but that have not been adequately explored in existing literature.	
From the background and technical information given in the extensive appendices, it is clear that great care	This is an accurate description of our effort – thank you.	-

was taken in the design and implementation of the study. Standard meta-analytic approaches, such as the “risk of bias” assessment for individual studies, had to be modified and methods developed to account for data overlap between studies.		
Some of the subgroup comparisons are interesting, though the labelling of Figs 3-5 is not clear (apparent duplicate studies not explained).	The apparent duplicate studies in Figures 3-5 represent different estimates of the weekend effect reported in same study - typically two separate estimates for Saturday vs weekday(s) and Sunday vs a weekday(s). We have added footnotes underneath these figures to explain this, and have also added text. During the revision we identified a number of coding inconsistencies in our data file, resulting in omission of some estimates (primarily Saturday vs weekday(s) from the analyses. We have re-checked the full dataset, rectified these errors and re-analysed the data, and have updated all relevant figures and estimates.	Manuscript p.11; supplementary file, Appendix 9, p.44-46, Figures 5-7 (please note that the Appendix and Figure numbers have been updated due to the inclusion of an extra appendix and additional figures in the supplementary file during the revision.
The inter-study variation in the ratio of numbers of weekend and weekday admissions and in the actual mortality rates shown in Table 7 is interesting. The apparently large weekend effect seen for elective admissions is bizarre and deserves fuller discussion. Among which type of elective admissions did the main mortality occur and what were the actual death rates? If they were admitted for procedures, were these done at weekends and when did the deaths occur? Were planned admissions for palliative care included? Surely this must be a case-mix (denominator) effect rather than reflecting any deficiencies in care – why would any hospital plan to bring patients in on days when it cannot provide adequate cover?	We agree with the reviewer that the apparently large weekend effect for elective admissions is likely attributable to case mix effect. Personal clinical experience and informal discussions with colleagues support the view that surgeons will often reserve surgery for multi-morbid or more complex patients until the weekend, as this allows longer time for the immediate post-operative recovery phase. We do not have sufficient information from individual studies to answer the important questions that the reviewer posed; some of the information (such as inclusion/exclusion of palliative care patients) was infrequently mentioned in published studies, and efforts to collect the detailed information warrant a separate review in its own right.	-
I have no substantial criticisms of this study – its only weaknesses are in the	Data from individual studies comparing the impact of different	

individual studies selected (systematically) for review. Minor changes might be made to the layout to help the reader (eg the order of topics in the Appendix differs from that in the Results section of the paper) and the labelling of some of the figures and tables could be clearer.	methodological features (such as different definitions of the weekend and different measures of mortality) on the estimated weekend effect were previously presented within the sensitivity analysis section of the appendices, and this differed from the order when they were mentioned in the Results section of the main text. We have now re-located them to be presented in a new appendix (Appendix 10) after the appendix on subgroup analysis so that the order is aligned with the main text. Numbering of appendices has been updated accordingly. We have improved the captions for figures and tables, and added footnote where necessary throughout the manuscript and the appendices in the supplementary file.	Appendix 10 Figures and tables throughout the manuscript and appendices
Although the authors emphasise that it is dangerous to draw conclusions from this mass of data about quality or optimal organisation of care, I feel they miss an opportunity to go further in their recommendations or future research. Instead of struggling to find ever more intricate methods of case-mix adjustment, why not mine the data to find plausible instances of deficiencies in care in specific conditions or situations, and possible ways in which they might be remedied? The more sophisticated case-mix, process and outcome data now becoming available routinely could then be used to mount cluster randomised trials (at minimal cost), which would eliminate much of the confounding that bedevils current inter-hospital and time trend comparisons. This study is a good illustration of the fact that, even by analysing huge numbers of cases, using sophisticated statistical and intensive computational techniques to reduce noise in the data, we will never produce a LIGO detector for health care effects...	We share this perspective and have added some texts (as described in one of our responses above) to highlight the potential for utilising data from routine databases to pin point areas for focused investigation. Indeed, the High Intensity Specialist Led Acute Care (HiSLAC) project funded by the HS&DR programme, of which this systematic review is part, is focused specifically on addressing the issue of causation. In our conclusions we state: "Future research and interpretation of research findings on the weekend effect must go beyond the narrow focus of case mix adjustment of routine hospital data and attempt to examine the broader issues related to the whole care pathway both within and outside the hospital..." As an example, we have recently published an analysis of four years of data from the Queen Elizabeth Hospital Birmingham:  • Sun J, Aldridge C, Girling A, Evison F, Beet C, Boyal A, Rudge G, Lilford R, Bion J, on 	Manuscript p.24

	behalf of the HiSLAC Collaboration. Sicker patients account for the weekend mortality effect amongst adult emergency admissions to a large hospital trust. BMJ Quality & Safety 2018 Epub ahead of print October 2018: doi:10.1136/bmjqs-2018-008219	
1 typo on p21, ln7 –“two-folds” (this sentence should be shortened)	We have corrected this typo and shortened the sentence.	Manuscript, p.23
Reviewer 3: Toshiya Shiga, International University of Health and Welfare, Japan		
Very comprehensive review with the elegant statistic methods.	Thank you.	-
I would like to mention the only one thing. The authors need to show the level of evidence with GRADE (Grading of Recommendations, Assessment, Development and Evaluation) system on each outcome (in this case, mortality, length of stay, or stuff like that)	The GRADE framework was originally developed for interpretation of evidence on intervention effectiveness for the purpose of guideline development, for which the target population and setting tend to be narrowly and strictly defined. This is not the scenario in our review, in which we are exploring evidence from a wide range of settings rather than focusing on a specific setting. Therefore full implementation of GRADE may not be practical. Nevertheless, we concur with the reviewer that the domains highlighted in GRADE (risk of bias, precision, inconsistency, indirectness and publication bias) are important and we took them into account when interpreting findings in our manuscript. For mortality outcomes, we devote specific sections on risk of bias assessment and publication bias and highlight the issue of heterogeneity and inconsistency in the Results section. For other outcomes, the studies are too heterogeneous in patient population and in outcome measures covered to allow rigorous assessment based on GRADE.	Manuscript, P 14
Reviewer 4: Jacob Simmering, University of Iowa, United States		

Chen and co-authors present a systematic review and meta-analysis of the literature regarding the "weekend effect" among inpatient admissions. They report finding an increased risk of inpatient mortality among those admitted on the weekend relative to other days of the week. They do not find any additional risk for adverse events for patients admitted during the weekend. The average and median length of stay among weekend admissions was somewhat (less than 1 day) shorter than during weekday admissions - an effect attributed to greater risk of inpatient mortality. In general, the review and meta-analysis seem complete and statistically well-grounded. I have some limited concerns about the authors interpretations of the regression model and about the possible sources of the weekend effect. My full comments are below.	Thank you. We address the reviewer's individual concerns below.	-
Statistical Analysis The use of Bayesian modeling is well-suited to this meta-analysis and provides easy to understand results in the form of posterior probabilities and a well-specified model. I have some brief comments here:	Thank you.	-
1. The convergence was assessed by visual inspection of the MCMC traces and the $R_{\hat{}}$ statistic. I would like to see, at least in an appendix, the MCMC traces and I would like to see the $R_{\hat{}}$ statistics and effective sample sizes (n_{eff}) reported in the main paper. These are important to assessing the quality of the convergence and the extent to which the posterior space has been sampled. Without being confident that the chains are well-mixed and converged, it is hard to assess whether the inference based on the sampling is valid.	We have now provided the Hamiltonian Monte Carlo (HMC) traces, $R_{\hat{}}$ statistic and effective sample sizes in the supplementary file (we feel these details would be too technical for general readers, but will be happy to follow editor's advice), and briefly mentioned these in the main text.	Supplementary file, Appendices 7 & 8
2. The MCMC was only run for 2,000 iterations. While convergence, especially in a relatively straightforward model can be achieved in such few runs, I would prefer to see a larger number of iterations (perhaps 10,000 or	We have accepted the reviewer's recommendation and have re-run each analysis for 10,000 iterations including 2,000 warm-up/burn-in iterations or more iterations in the same proportion if convergence was judged not to have been achieved.	Manuscript p.11

more) to be sure that any low-probability area in the posterior density is explored.	These have produced results are very similar to those reported in our original submission. We have nonetheless updated relevant numbers and figures in this article.	
3. What was the number of iterations discarded as burn-in? What is the number of iterations used for inference?	As described above, we have now run each analysis for at least 10,000 iterations including 2,000 warm-up/burn-in iterations or more iterations in the same proportion if convergence was judged not to have been achieved.	Manuscript, p.11
4. The authors select weak priors but opt for a normal distribution (or half-normal distribution). Can you motivate these priors over alternatives (such as $U(0, \infty)$ or $U(-\infty, \infty)$)? I don't think this would change the results - the selected standard deviations are \gg the estimated size of the effect. It would be nice to provide some basis for selecting the standard normal priors based the a priori estimated size of the parameter.	"Weakly informative" priors were chosen over alternatives since these restrict the parameter space to a plausible range and provide a degree of regularisation. This facilitates computation especially with small numbers of studies, but provides relatively little information within the plausible range. For example, a $N(0,1)[0, \infty)$ prior for between study heterogeneity has a 95th percentile of 1.96, which would be considered large given a within-study estimated standard deviation for the weekend effect of between approximately 0.01 and 0.05. Previous research also suggests higher level variance terms in meta-analysis rarely exceed 0.2 in these contexts (see Turner et al. 2015 https://doi.org/10.1002/sim.6381). In contrast, the $U(-\infty, \infty)$ suggested priors are improper and are unlikely to produce an interpretable posterior, especially in analyses with lower numbers of studies. Indeed, the priors would suggest that the variance terms are exceedingly more likely to be >10 than <10, which is implausible in the extreme.	Supplementary file Appendix 3.2
Statistical Interpretation 1. On page 15, the authors report that the estimated weekend effect diminished with increased statistical control. Specifically, they reported that studies with "adequate adjustment of main and contextual factors" had a 14% greater estimated weekend effect than studies with "adjustment including measures of acute physiology." I am concerned by this interpretation are no	Like the reviewer, we had hypothesized that the estimated weekend effect might be associated with adequacy of statistical adjustment as measured by our comparability rating. The meta-regression results turned out to show no discernible differences between estimates with comparability ratings 2b, 3 and 4. Nevertheless, there	Manuscript p.17

further dose-response pattern is observed. If we are to understand the difference between studies with control levels of 1/2a and 2b as being caused by insufficient statistical control, we would expect a similar effect moving to partial adjustment (3) and inadequate adjustment (4). However, we do not observe such an effect. I am concerned that the observed difference between control levels 1/2a may have more to do with study design (e.g., access to labs data) than with the adequacy of statistical control.	appear to be a difference in the expected direction between estimates with comparability ratings 1 and 2a and other estimates. As the main distinction between studies with 1 and 2a rating and the rest of studies was the adjustment of measures of acute physiology, we believe our statement is a fair description of the findings of the meta-regression, and disagree that it is a misinterpretation of the evidence. The lack of dose-response pattern among category 2b, 3 and 4 could indicate the inadequate reliability of the subjective, ordinal comparability rating, but it could also reflect a 'threshold effect' (i.e. adjustment is inadequate unless measures of acute physiology is included). We appreciate the reviewer for pointing out the possibility of confounding through differential accessibility of laboratory data, and have added the following text in Point 4 of our statement which cautions about the interpretation of the findings: "Therefore there is still a substantial level of uncertainty, and the apparent effect of adjustment of acute physiology could have been confounded by other patient or service features associated with the availability of these measures."	
2. Moreover, the reported effects are not statistically significant and the reference group strata is sparse (n = 5). The authors acknowledge this in point 4 on page 15; however, this is not in agreement with the initial claims in point 1. Point 1 seems to substantially overstate the evidence provided by this analysis.	As we are adopting a Bayesian approach, we place more emphasis on congruence between our prior belief (which was informed by theories and prior literature) and observed data rather than on statistical significance. We feel that we have been careful in providing a fair account of the findings, and have modified Point 4 as described above to provide a stronger warning regarding relevant caveats.	Manuscript p.17
Major Comments 1. Figure 2 seems unsupported by the text. This perhaps belongs in the sister paper. This paper seems simply about the presence and size of the weekend effect and does not seem to explore the possible causes.	Figure 2 is a distillation of thoughts on issues that ought to be considered when the weekend mortality effect is examined with an intention to infer the quality of hospital care during weekends compared with weekdays.	

Likewise, the sentence in the conclusion that the weekend effect “may be largely attributable to case mix and contextual factors surrounding admissions, and therefore the cause may lie upstream of the care pathway, in the community” seems unsupported by the analysis and review presented.	We did not claim that such a “mind map” is derived from data presented in this article, and indeed have presented the Figure in the Discussion section rather than the Results section. We feel this information is important to highlight how a vast volume of current literature might have focused too narrowly on a small number of determinants of weekend/weekday mortality (for which data are readily available) without considering other important factors such as differential availability services outside hospital and data quality between weekends and weekdays. The figure would be useful to help readers to put the findings into context, and can also assist future researchers to develop logic models, the importance of which are increasingly being recognised for the evaluation of service delivery and organisation within the complex health system. Our conclusion that the weekend effect may be attributable to factors upstream of admission is based on the fact that since patients are sicker at hospital presentation at weekends, then there must be some cause in the pre-admission phase.	
2. While it can be argued that the weekend effect is not a measure of quality of care (while mortality is increased, adverse events are not suggesting the observed increase in mortality may be the result of case-mix changes and not changes in care quality), this paper does not seem to strongly argue that position. This section of the results and conclusion seem to overstate the strength of the evidence presented in the paper.	The reviewer correctly identifies the tension between strength of evidence and strength of recommendations. We cannot argue strongly that in-hospital quality of care is as good at weekends as on weekdays (though there is now some evidence that it may be better at weekends for emergency admissions), but we can argue that weekend admissions are sicker at hospital presentation. We cannot therefore absolve hospitals from contributing to the weekend effect, but we can be very clear that weekend community services require examination.	-
3. The discussion of the companion paper seems out-of-place in this paper. This paper’s focus, as I understand it, is	We have removed relevant texts from the Abstract, Methods and Discussion.	Manuscript p.4, p.13 and p.25

a review and meta-analysis of the literature regarding the weekend effect on mortality, adverse events and length-of-stay. The companion paper explores possible causes of this effect. It is unclear why the companion paper is being discussed in this review.		
Minor Comments 1. The Figure 1 includes the posterior predictive estimate and credible interval; however, this value is never referenced in the text. Please discuss this value in the text.	We have now included the following sentences in the main text: “Posterior predictive interval suggests that if a new study were to be undertaken, the estimated odds ratio is likely to lie somewhere between 1 (no weekend effect) and 1.34 (the odds of death being 34% higher for weekend admissions compared with weekday admissions).”	Manuscript p.16
2. The paper’s hierarchy is somewhat hard to follow. The transition from sensitivity analyses to the results on adverse effects/LOS/patient satisfaction was somewhat surprising. The headings do not indicate hierarchy well or there are too many levels of hierarchy.	We acknowledge it is very challenging to organise and present the large volume of information from this complex systematic review in a simple way. Our pre-specified focus on mortality outcomes over other outcomes (and correspondingly the differential sophistication of analyses carried out for different outcomes) created the asymmetry between the section on mortality outcomes and subsequent sections for other outcomes. Nevertheless we did try to maintain a consistent heading style which reflects the hierarchy. We have now added the following texts to the beginning of Data synthesis sub-section of the Methods section to prime readers about the asymmetry: “Our pre-specified primary outcome is mortality. The quantitative synthesis methods described below were used to analyse mortality data, which form the main part of this article. Data related to adverse events, length of hospital stay and patient satisfaction were tabulated and presented in the supplementary file, with a brief narrative summary provided in this article.” Given the constraint on resources, it is not possible for us to undertake	Manuscript p.10

	further analyses for non-mortality outcomes.	
--	--	--

VERSION 2 – REVIEW

REVIEWER	Elizabeth Kuhn University of Alabama at Birmingham United States of America
REVIEW RETURNED	28-Jan-2019

GENERAL COMMENTS	The authors present a revised manuscript describing the results of their systematic review and meta-analysis of studies concerning the weekend effect. The review focuses only on hospital-wide samples (as opposed to condition-specific studies). The review includes up-to-date publications through November 2017. Page 10 – the ad hoc risk adjustment categories “based on emerging evidence on key confounding factors” should cite the evidence they relied on in creating those categories. I have a hard time considering care delivered from 1985 – 2000 to be “contemporary.” For the purposes of this study, it’s hard to conceive that those are sufficiently similar populations and healthcare processes to make comparison meaningful. The authors should exclude studies incorporating data from prior to year 2000 (or 1997 which would be a 20 year span). Reviewing Appendix 6, this would mean excluding: Madsen 2014, Gordon 2005, Bell 2001, Goldstein 2014, Gould 2003, Luo 2004, Pasupathy 2010, and Salihu 2012. The Gordon paper includes patients admitted to the VA from 1991 to 1993, Bell from 1988-1997, Gould 1995-1997, Luo from 1985-1998, and Salihu from 1989-1997. Data from these time periods cannot be meaningfully compared to data within the last 20 years due to substantial changes in healthcare organization and processes. The included study that was based on cross-sectional surveys should be described in more detail. It is a littler unclear to me how weekend effect data would be gathered using this study design. There should be more discussion of elective admissions, since that is the subgroup that saw the greatest magnitude of a weekend effect. Specifically, there should be discussion of the characteristics of elective admissions that are occurring on weekends—what hospitals/countries, what diagnoses, etc. In North America, this is generally an uncommon practice and factors that lead to elective weekend admissions may be confounders (increased “urgency” of elective procedure, inadequate staffing or bed capacity during week – “overflow”, etc.).
--

REVIEWER	David Barer Newcastle University, UK
REVIEW RETURNED	24-Jan-2019

GENERAL COMMENTS	The authors have responded in detail to all the points raised, relabelled the figures, substantially reorganised the supplementary appendices and added a lot of new material, so that the whole file now runs to 167 pages. They have even done a lot of further analysis, running 8,000 more iterations of the Bayesian model to check convergence in all subgroups. Unfortunately it looks as though many of the reference numbers given in the tables (especially Appendices 6 and 10) may need correcting. I have not checked through all of the other changes but am entirely satisfied with the responses to the points I raised. I also strongly support their response to the statistical reviewer, who disputed their interpretation of the relationship between estimates of the weekend effect in different studies and the adequacy of adjustment for confounding. Apart from distinguishing between emergency and elective admissions, administrative databases contain little information likely to confound the weekend mortality effect to any extent (some major datasets cannot even identify out of hours admissions), so it is not surprising that no clear ordinal relationship was seen across all 5 "levels of statistical control". To clinicians it seems obvious that there will be a higher threshold for emergency admission at weekends, so we would expect the most important difference between unplanned weekday and weekend admissions to be in the degree of illness severity - and very few studies were able to adjust for physiological factors. Given the glaring deficiencies in the ability of most published studies to deal with confounding, I feel the authors' overall conclusions are very conservative. In fact I dread seeing more headlines about excess hospital deaths at weekends, based on selective reporting of figures from the abstract. The authors should be congratulated on completing a huge amount of work, not only in extracting and analysing all the data, but in developing new methods for assessing and combining complex, heterogeneous and overlapping studies. Normally the use of ad hoc methods for a particular study should be discouraged, but in this case I am sure they will prove useful for future research. I also thank them for pointing out the recent BMJQS paper, analysing emergency admissions in Birmingham, which confirms that the apparent weekend mortality effect can be all but eliminated by efficient adjustment for illness severity. Clear conclusions cannot be drawn without knowing the out-of-hospital death rates, but given that similar numbers of patients present to A&E at weekends, while fewer are admitted, the findings might simply indicate that the emergency hospital admissions system is working more efficiently at weekends...
---

REVIEWER	TOSHIYA SHIGA Department of Anesthesiology and Intensive Care Medicine, International University of Health and Welfare,
REVIEW RETURNED	22-Jan-2019

GENERAL COMMENTS	The GRADE framework can be implemented in not only meta-analysis of RCT, but meta-analysis of observational studies such as yours. Please go onto the following HP https://gradepro.org/ and assess your outcome. You need to look up how score will upgrade. If your study will turn out to be "low", its not a problem since your study targets observational study. Observational study tends to be "low" in GRADE rating system.
---

REVIEWER	Jacob Simmering University of Iowa
REVIEW RETURNED	23-Jan-2019

GENERAL COMMENTS	The authors have extended the sampling of their MCMC chains and now report convergence diagnostics. The traces of the MCMC and the values for R_hat suggest that the space was reasonable well-explored and the chains are well mixed. The reporting in the main text is brief but probably sufficient for the audience of BMJ Open. I feel the discussion of the use of weakly informative priors is reasonable. I am still concerned about overstating the effect of adequate adjustment for confounding between groups 1/2a and the remaining groups but the revised statement in point 4 adequately raises these concerns. I would like to see the credible interval for the 15% claim in point (1) added inline as the credible interval does include the null. The authors could also add the posterior probability of the estimates from studies with adjustment levels 2b, 3, 4 being > the estimates for adjustment level 1/2a.
---

VERSION 2 – AUTHOR RESPONSE

Authors' responses to reviewers' comments

Reviewers' comments to authors	Authors' responses & changes made (if applicable)	Page number (in revised, track-changed, manuscript)
Reviewer 1: Elizabeth N Kuhn, University of Alabama at Birmingham		
The authors present a revised manuscript describing the results of their systematic review and meta-	We have now provided the references.	p.10

analysis of studies concerning the weekend effect. The review focuses only on hospital-wide samples (as opposed to condition-specific studies). The review includes up-to-date publications through November 2017. Page 10 – the ad hoc risk adjustment categories “based on emerging evidence on key confounding factors” should cite the evidence they relied on in creating those categories.		
I have a hard time considering care delivered from 1985 – 2000 to be “contemporary.” For the purposes of this study, it’s hard to conceive that those are sufficiently similar populations and healthcare processes to make comparison meaningful. The authors should exclude studies incorporating data from prior to year 2000 (or 1997 which would be a 20 year span). Reviewing Appendix 6, this would mean excluding: Madsen 2014, Gordon 2005, Bell 2001, Goldstein 2014, Gould 2003, Luo 2004, Pasupathy 2010, and Salihu 2012. The Gordon paper includes patients admitted to the VA from 1991 to 1993, Bell from 1988-1997, Gould 1995-1997, Luo from 1985-1998, and Salihu from 1989-1997. Data from these time periods cannot be meaningfully compared to data within the last 20 years due to substantial changes in healthcare organization and processes.	We respectfully disagree. Health services are evolving the whole time and there is no specific reason why studies with admissions pre-2000 should be removed any more than those pre-2005 etc. Importantly, the comparisons inherent in this review concern the weekend effect (i.e. difference in outcomes between weekend and weekday admissions estimated within the study period of respective studies) rather than a direct comparison of services or outcomes between different time periods. This difference-in-difference analysis takes into account within-period changes in practice, while comparison of different time periods allows exploration (as we did through meta-regression and examination of data reported within individual studies) to determine whether the weekend effect changes overtime given the continuous change/improvement in health services. This provides further insight for our understanding of the weekend effect – for examine, a fairly constant weekend effect amid known changes in hospital services would lend support to the hypothesis that the weekend effect is not caused by hospital service issues. We therefore stand by the approach that we have adopted with regard to study year.	-
The included study that was based on cross-sectional surveys should be	We have revised the sentence to read: “All studies were retrospective	p.14

described in more detail. It is a little unclear to me how weekend effect data would be gathered using this study design.	cohort studies except one which was based on comparison of responses to cross-sectional surveys between patients admitted at weekends and those admitted during weekdays.”	
There should be more discussion of elective admissions, since that is the subgroup that saw the greatest magnitude of a weekend effect. Specifically, there should be discussion of the characteristics of elective admissions that are occurring on weekends—what hospitals/countries, what diagnoses, etc. In North America, this is generally an uncommon practice and factors that lead to elective weekend admissions may be confounders (increased “urgency” of elective procedure, inadequate staffing or bed capacity during week – “overflow”, etc.).	Thank you. We have revised the relevant paragraph in the Discussion to read: “The finding that weekend mortality effect is larger among elective than emergency admissions might be explained by a greater case mix difference between weekends and weekdays that is unaccounted for by statistical adjustment among elective admissions compared with emergency admissions. For example, for procedures that are often carried out during elective admissions, such as hip and knee replacement and surgery for large bowel, switching the timing of admission from weekdays to weekends due to change in urgency (which is unlikely to be captured by administrative database) or delay in admission during weekdays due to capacity issues (‘overflow’) is fairly plausible. On the other hand, a greater weekend effect associated with elective admissions is also consistent with the hypothesis that hospitals are configured to care for emergencies at weekends, while elective admissions might be overlooked.”	p.25.
Reviewer 2: David Barer, Newcastle University, UK		
The authors have responded in detail to all the points raised, relabelled the figures, substantially reorganised the supplementary appendices and added a lot of new material, so that the whole file now runs to 167 pages. They have even done a lot of further analysis, running 8,000 more iterations of the Bayesian model to check convergence in all subgroups.	Thank you for recognising our substantial effort.	-
Unfortunately it looks as though many of the reference numbers given in the tables (especially	We have inspected the tables and could not identify any errors. We suspect a potential confusion might have arisen from the fact that we	Supplementary file, p.1.

Appendices 6 and 10) may need correcting.	have constructed a separate reference list for the appendices and therefore the reference numbers cited in the appendices are different from those cited in the main manuscript. We have added the following note to the covering page of the supplementary file (appendices) to draw readers' attention to this: "Please note that references cited in this supplementary file are listed at the end of this document. Therefore the reference numbers cited in-text correspond to reference numbers of the reference list within this supplementary file, and they are different from the reference numbers quoted in the main paper."	
I have not checked through all of the other changes but am entirely satisfied with the responses to the points I raised. I also strongly support their response to the statistical reviewer, who disputed their interpretation of the relationship between estimates of the weekend effect in different studies and the adequacy of adjustment for confounding. Apart from distinguishing between emergency and elective admissions, administrative databases contain little information likely to confound the weekend mortality effect to any extent (some major datasets cannot even identify out of hours admissions), so it is not surprising that no clear ordinal relationship was seen across all 5 "levels of statistical control". To clinicians it seems obvious that there will be a higher threshold for emergency admission at weekends, so we would expect the most important difference between unplanned weekday and weekend admissions to be in the degree of illness severity - and very few studies were able to adjust for physiological factors.	We thank the reviewer's support for our interpretation of the evidence.	-

Given the glaring deficiencies in the ability of most published studies to deal with confounding, I feel the authors' overall conclusions are very conservative. In fact I dread seeing more headlines about excess hospital deaths at weekends, based on selective reporting of figures from the abstract.	We concur and believe the conclusion that we offer is as objective as possible and can re-direct attention of researchers and the public from potentially unreliable numerical estimates to understanding the underlying mechanisms of these estimates.	-
The authors should be congratulated on completing a huge amount of work, not only in extracting and analysing all the data, but in developing new methods for assessing and combining complex, heterogeneous and overlapping studies. Normally the use of ad hoc methods for a particular study should be discouraged, but in this case I am sure they will prove useful for future research.	Thank you.	-
I also thank them for pointing out the recent BMJQS paper, analysing emergency admissions in Birmingham, which confirms that the apparent weekend mortality effect can be all but eliminated by efficient adjustment for illness severity. Clear conclusions cannot be drawn without knowing the out-of-hospital death rates, but given that similar numbers of patients present to A&E at weekends, while fewer are admitted, the findings might simply indicate that the emergency hospital admissions system is working more efficiently at weekends...	We agree.	-
Reviewer 3: Toshiya Shiga, International University of Health and Welfare, Japan		
The GRADE framework can be implemented in not only meta-analysis of RCT, but meta-analysis of observational studies such as yours. Please go onto the following HP https://gradepro.org/ and assess your outcome. You need to look up how score will upgrade. If your study will turn out to be "low", its not a problem since your study targets observational study.	Thank you for the information and advice. We have assessed the overall quality of evidence for the four key outcomes examined in our review based on the GRADE framework. Evidence for all four outcomes was judged to be of 'very low' quality/certainty primarily due to the low baseline rating associated with observational studies and the inadequate adjustment of potential confounding factors. We have added	p.4 Abstract; p.13 Methods; p. 23 Results; p.25 Discussions; p.28 Conclusion; Supplementary file Appendix 14

Observational study tends to be "low" in GRADE rating system.	relevant information in various parts of the main text, and have included the justification for our GRADE assessment in Appendix 14.	
Reviewer 4: Jacob Simmering, University of Iowa, United States		
The authors have extended the sampling of their MCMC chains and now report convergence diagnostics. The traces of the MCMC and the values for R_hat suggest that the space was reasonable well-explored and the chains are well mixed. The reporting in the main text is brief but probably sufficient for the audience of BMJ Open.	Thank you.	-
I feel the discussion of the use of weakly informative priors is reasonable.	Thank you.	-
I am still concerned about overstating the effect of adequate adjustment for confounding between groups 1/2a and the remaining groups but the revised statement in point 4 adequately raises these concerns.	Thank you.	-
I would like to see the credible interval for the 15% claim in point (1) added inline as the credible interval does include the null. The authors could also add the posterior probability of the estimates from studies with adjustment levels 2b, 3, 4 being > the estimates for adjustment level 1/2a.	We feel that the current text for point (1) is the most concise and clear statement. As there are three estimates (for group 1/2a compared with group 2b, 3 and 4 respectively) all showing very similar figures, it would become unwieldy and somewhat repetitive if we quote each of the (slightly different) estimates with their credible intervals in-line. Readers are referred to Table 1 where these figures are shown at the beginning of this paragraph, and we have now also added the following text in point (4): "...and the 95% credible intervals include zero (Table 1)" to highlight the point that the reviewer raised. From a Bayesian perspective, we feel this would be sufficient as we are not carrying out any hypothesis testing and we would not overly emphasise the arbitrary boundary defined by 95% credible intervals.	p.17-18

VERSION 3 - REVIEW

REVIEWER	Elizabeth N (Kuhn) Alford University of Alabama at Birmingham, USA
REVIEW RETURNED	20-Mar-2019

GENERAL COMMENTS	I congratulate the authors not only on their comprehensive and exhaustive work, but their attention to detail in revising their manuscript. It is very well written and constitutes an important contribution to the existing literature. I am eager to see the framework synthesis from Part 2 of this project. They have responded appropriately to the concerns of the statistical reviewer. Overall their statistical approach is elegant and well-described.
---

REVIEWER	Toshiya Shiga Department of Anesthesiology, International University of Health and Welfare, Japan
REVIEW RETURNED	18-Mar-2019

GENERAL COMMENTS	Lovely!
---------